

# Recent advances in methods for *in situ* root phenotyping

Anchang Li[1,*], Lingxiao Zhu[2,*], Wenjun Xu[1], Liantao Liu[2] and Guifa Teng[1]

[1] School of Information Science and Technology, Hebei Agricultrual University, Baoding, Hebei, China
[2] State Key Laboratory of North China Crop Improvement and Regulation, Hebei Agricultrual University, Baoding, Hebei, China
* These authors contributed equally to this work.

## ABSTRACT

Roots assist plants in absorbing water and nutrients from soil. Thus, they are vital to the survival of nearly all land plants, considering that plants cannot move to seek optimal environmental conditions. Crop species with optimal root system are essential for future food security and key to improving agricultural productivity and sustainability. Root systems can be improved and bred to acquire soil resources efficiently and effectively. This can also reduce adverse environmental impacts by decreasing the need for fertilization and fresh water. Therefore, there is a need to improve and breed crop cultivars with favorable root system. However, the lack of high-throughput root phenotyping tools for characterizing root traits *in situ* is a barrier to breeding for root system improvement. In recent years, many breakthroughs in the measurement and analysis of roots in a root system have been made. Here, we describe the major advances in root image acquisition and analysis technologies and summarize the advantages and disadvantages of each method. Furthermore, we look forward to the future development direction and trend of root phenotyping methods. This review aims to aid researchers in choosing a more appropriate method for improving the root system.

## INTRODUCTION

Grain yield in developing countries increased by 208% between 1960 and 2000, attributed to the first Green Revolution, which led to the development of semi-dwarf wheat and rice varieties (*Pingali, 2012*). However, the green revolution has been associated with many adverse effects, including the overuse of fertilizers and pesticides and soil degradation. Furthermore, mineral-based fertilizers like phosphorus are non-renewable resources that take between 80 to 100 years to deplete (*Isherwood, 2000*). Meanwhile, the efficiencies of nitrogen, phosphorus, and potassium fertilizer are ≤50% (*Elliott et al., 2018*), ≤30% (*Joseph, Soon & Ashley, 2016*), and ≤60% (*Zhu et al., 2017*), respectively. This is even more challenging given the impact of climate change on water availability and efforts to reduce fertilizer inputs to ensure environmentally friendly and sustainable agriculture (*Atkinson et al., 2018*). Therefore, there is a need to develop crops with improved water and

Corresponding authors
Liantao Liu, liultday@126.com
Guifa Teng, tguifa@hebau.edu.cn

nutrient uptake efficiency, which is the main aim of the second Green Revolution (*Lynch, 2007*; *Lynch, 2022*).

Roots absorb water and nutrients from soil and are vital to the survival of nearly all land plants, especially because plants are anchored and cannot move to find more favorable growing conditions. Root phenotype has an important relationship with crop water and nutrient uptake and greatly affects shoot development and yield formation. Therefore, improving root traits is a key target for the second Green Revolution. Root phenotype is controlled by the coordination between intrinsic genetic factors and external environmental conditions (*Lynch, 1995*; *Malamy, 2010*) and is a key element of yield improvement. The root system facilitates a series of adaptive responses at the cellular and organ level under unfavorable external environment (*Miroslaw et al., 2016*) and ensures a high level of plasticity (*Gruber et al., 2013*). Root plasticity is the prerequisite for genetic improvement of root traits and a key element of yield improvement. *Tracy et al. (2020)* summarized 11 programs that integrated root traits into germplasm for breeding by phenotyping. However, the development of root phenotyping techniques, especially *in situ* root phenotyping has lagged behind due to hidden nature of the root structure in the soil and the high complexity of the root system (*Lynch, 2021*; *Delory et al., 2022*). There is an urgent need to establish accurate and efficient root phenotyping technologies for measuring root properties, including root system architecture and morphology under various stresses (*McCormack et al., 2017*).

Traditional root phenotyping methods, such as soil core, trench, mesh bag, shovelomics, and monolith, are all destructive, since they involve isolating the root system from the soil to obtain the root topology and phenotype. The soil core method, which is the most common technique for assessing the root system, entails obtaining rooted soil blocks from the field, washing, and selecting the root system components (*Kücke, Schmid & Spiess, 1995*). Thus, this method only obtains partial data of the root system due to limited sample collection and difficulty in obtaining the root system of a single plant (*Takahashi & Pradal, 2021*). The trench method is one of the earliest and most used root research methods, involving excavating the soil at a certain distance and depth from the plant and then washing out the roots (*Livingston, 1922*). However, the trench method is time-consuming and labor-intensive (*Takahashi & Pradal, 2021*). The mesh bag method involves digging a hole of a certain diameter in the field, putting a mesh bag into the hole, and backfilling the soil; the mesh bag is then taken out with the roots which are then washed (*Steen, 1991*). Nevertheless, this method is limited to the study of roots in the upper soil, because it is difficult to obtain the mesh bag in the later stage. The shovelomics has enabled high-throughput root phenotyping of field grown crops, where 20 cm of root material immediately below the surface is excavated, washed, and imaged (*Trachsel et al., 2011*). The above-mentioned root sampling methods have been gradually improved to facilitate the research in root phenotyping; However, their destructive sampling techniques often result in finer-scale root features being lost (*e.g.*, finer lateral roots and root hair) and only a snapshot of development being measured (*Bucksch et al., 2014*). More importantly, destructive sampling methods are time-consuming and labor-intensive, with a high root

loss rate. Thus, there has been a need to develop faster and more accurate methods for *in-situ* observation of root phenotype.

Non-invasive and high-throughput root phenotype analysis methods are essential for studying root phenotype and its change dynamics. Novel techniques are needed to automatically describe the complexity of the root system and identify root phenotype traits. At present, the acquisition and analysis methods of *in situ* root system are still in the development stage. However, no comprehensive review is available on the *in situ* root phenotyping methods and image processing software. Hence, we summarize the advances in research methods of *in-situ* root system analysis from two aspects: *in-situ* root cultivation and imaging system and image processing software. In addition, the cutting-edge technology of *in-situ* root system observation is summarized and analyzed to provide reference for plant root system research. This article should be of particular interest to readers in the areas of plant morphology, especially root morphology, and related platform and software development.

## SURVEY METHODOLOGY

Literature was collected as previously described in *Amoanimaa-Dede et al. (2022)*. Specifically, primary and secondary literature relevant to this review was accessed using Web of Science and Google scholar. Key words such as "root phenotyping", "*in situ*", "root morphology", "platform" and "software" were searched between 22 February and 15 March, 2022. Although there are some reviews on root phenotyping technology like *Tripathi et al. (2021)*, *Tracy et al. (2020)*, and *McGrail, Van Sanford & McNear (2020)*. However, there has not been a comprehensive review on *in situ* root phenotyping method. Meanwhile, we mainly focused on works from the past 15 years. Literature was retrieved and sorted based on the relevance of the topic. Together, the compiled information was processed by the authors to write the manuscript. Relevant methods and software were incorporated based on the author's expertise in this field of research.

### 2D root phenotyping platform

The most widely used method for root phenotyping is the 2D root phenotyping platform (*Delory et al., 2022*). This method consists of a growth system, imaging device, and image processing software. Here we divide the 2D root phenotyping method into two categories based on the culture medium: soil and soil-less culture methods (Table 1) (Fig. 1).

Growing plants in soil-less medium allow clear visualization of roots from the background and high-throughput control of environment for treatment evaluation (*Ana et al., 2015*). Soil-free methods include aeroponics, hydroponics, pouch-and-wick system, and agar gel-based phenotyping systems (*Kuijken et al., 2015*). An aeroponic system was proposed by *Carter (1942)*. The aeroponic system consists of air compressor, water pump, and incubator. Notably, the composition of air, nutrient solution, and ejection pressure in the aeroponics system can be adjusted as required (*Soto, 1982*). Aeroponics is mainly used to study the root structure of vegetables (*Tiwari et al., 2020*). Hydroponics is a high-throughput phenotype screening and identification method which

**Table 1 Overview of currently available root image analysis software advantages/limitations of root phenotyping methods and technologies.**

| Dimension | Medium | Advantages | Limitations | Examples | References |
|---|---|---|---|---|---|
| 2D | Aeroponics/ Hydroponics/ Pouch-and-wick system/agar | Providing a strong contrast between the root and background/Short period/ High-throughput /Allow-ing accurate extraction of root system architecture | Limited representation of actual root characteristics/Usually used in seedling stage/ Not suitable for studying root hairs | RhizoTubes/ Rhizoponics/Rhizoslides/ RhizoChamber-Monitor/ PlaRom/ChronoRoot | *Jeudy et al. (2016) Mathieu et al. (2015) Marié et al. (2014) Wu et al. (2018) Yazdanbakhsh & Fisahn (2009) Gaggion et al. (2021)* |
| | Soil | Allowing long-term observation/Close to the field conditions | Soil heterogeneity augments environmental noise/Root segmentation is relatively difficult/ Relatively low resolution | RhizoPot/ GROWSCREENRhizo/ GLO-Roots/GLO-Bot/ PhenoRoots/WinRoots | *Xiao et al. (2020) Treurnicht, Pagel & Esler (2015) Rubén et al. (2015) LaRue et al. (2021) Martins et al. (2020) Zhang et al. (2021a)* |
| 3D | Soil | Visualizing the dynamic development of complete root systems in natural soils/ Generating spatial and time resolved data | Low-throughput/High startup cost/Difficulty resolving fine roots duo to relatively Low-throughput | X-ray computed tomography/Magnetic resonance imaging/ Ground penetrating radar/Electrical resistivity tomography | *Heeraman, Hopmans & Clausnitzer (1997) van Dusschoten et al. (2016) Alnuaimy et al. (2000) Rossi et al. (2011)* |

involves culturing plants in a solid support device containing a nutrient solution with essential nutrients for plant growth. Hydroponic phenotyping system has been used to characterize root morphological traits at the early growth stage of various crop species, including soybean (*Chen, 2021*; *Salim et al., 2021*), barley (*Wang et al., 2021*), wheat (*Jeudy et al., 2016*; *Rumesh et al., 2019*; *Chen et al., 2020*), and maize (*Qiao et al., 2019*). *Jeudy et al. (2016)* developed a new tool for high throughput imaging of root features based on a form of hydroponic called RhizoTubes. The platform allows growing six plants simultaneously, and consists of an imaging cabin (Rhizo-Cab) that can automatically and non-destructively image both shoot and root compartments. However, this method has two drawbacks: first, hydroponics is not suitable for studying root hairs traits because it is uncertain whether root hairs can be formed in hydroponics environment. Second, hydroponics is only suitable for short-term root observation. As such, *Mathieu et al. (2015)* developed Rhizoponics tailored to characterize the root system of *Arabidopsis thaliana* from the seedling to adult stage. The pouch-and-wick system is an *in situ* observation system for roots based on germination paper. The method is affordable and simple to operate, and can be used to evaluate root morphology with high efficiency. It can also perform many repetitions and involves selecting a custom-colored germination paper that creates high contrast with root color to facilitate root image analysis. *Adu et al. (2014)* developed a low-cost, high-resolution, and simple root phenotyping platform based on pouch-and-wick system adaptable to most laboratories and glasshouses. Rhizoslides (*Marié et al., 2014*) and RhizoChamber-Monitor (*Wu et al., 2018*) are non-destructive and high-throughput root phenotyping platforms based on pouch-and-wick system. However, the main disadvantage of the pouch-and-wick system is that it can be only used to examine the root system of seedlings (*Hund, Trachsel & Stamp, 2009*). *Bengough et al. (2004)* proposed a root phenotyping method based on agar chamber to measure seedling

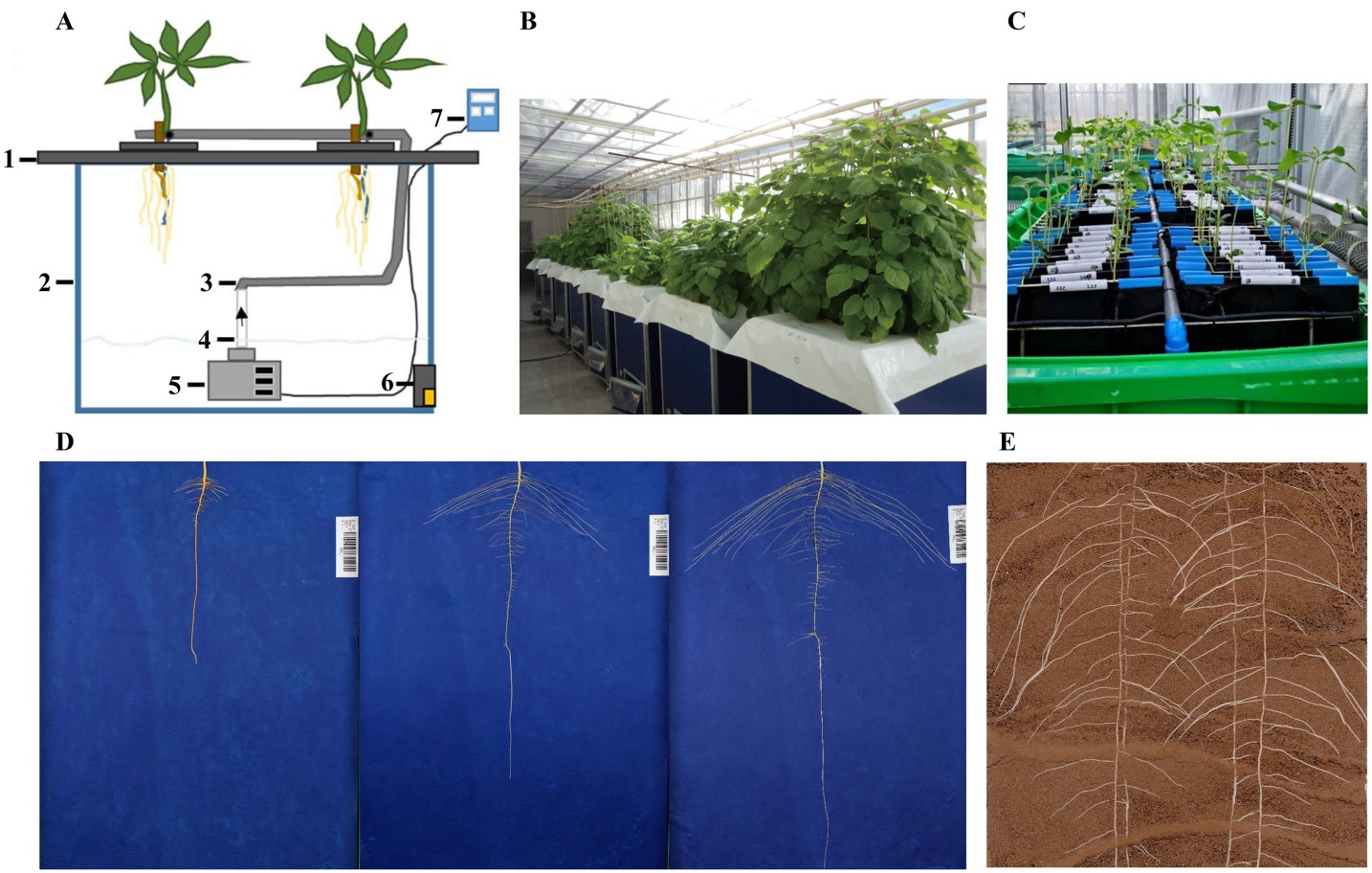

**Figure 1 Schematic representation of different 2D root phenotyping methods.** (A) Aeroponics (*Michael et al., 2019*); (B) aeroponics (*Jagesh et al., 2020*); (C) hydroponics (*Liu et al., 2021*); (D) pouch-and-wick system (*Kevin et al., 2020*); (E) soil-based (provided by our laboratory). 1. foam yumbolon; 2. container; 3. flexible drip hose; 4. PVC tube; 5. submersible pump; 6. drain valve; 7. timer.

root traits. The method involves growing seedlings between two closely spaced flat layers containing transparent gel. Subsequently, the root system traits are non-destructively recorded by a flatbed scanner. Root length, elongation rate, seminal root number, and other root traits can be easily obtained using this method. It is noteworthy that root growth in the gel chambers is very similar to that in the loosely packed soil, and is comparable to root growth of wild, landrace, and cultivated barleys in loosely packed soil. *Yazdanbakhsh & Fisahn (2009)* developed a high throughput platform for root hair monitoring called PlaRom. This platform is effective in phenotyping root growth dynamics, lateral root formation, and root architecture. It consists of an imaging platform and root development profiling software. *Gaggion et al. (2021)* developed a high temporal resolution for phenotyping root system called ChronoRoot, allowing a comprehensive characterization of root growth dynamics. However, like the agar gel-based phenotyping systems, ChronoRoot is only suitable for studying the roots of seedlings due to the influence of gel system nutrient supply and support capacity. Notably, root traits of seedlings are not always representative of mature plants but may be a good predictor of

later developmental stage morphometry (*Tuberosa et al., 2002*; *Mcphee, 2005*). The inherent disadvantage of soil-less systems is their limited representation of actual root characteristics of plants grown in soils (*Cai et al., 2015*; *Kuijken et al., 2015*).

Root phenotyping platforms based on soil culture mostly involve planting plants in containers containing one or more transparent planes and using image acquisition devices to obtain root images *in situ*. Many soil culture-based root phenotyping platforms have been developed. For example, *Hammac et al. (2011)* developed a novel and low-cost approach for observing root hair development of oilseed species in response to water availability. This platform can track the development of a single root or root hair over short time intervals (less than 10 min). Similarly, RhizoPot is an *in-situ* root observation platform with a resolution of up to 4,800 dpi. In addition to obtaining some basic indicators of the root system status, the method can be used to study the morphology and lifespan of fine roots and root hairs (*Xiao et al., 2020*; *Zhang et al., 2021b*; *Zhu et al., 2022*). However, the above two platforms are disadvantaged by the limited depth of the culture vessel, which may affect the natural growth of the root system. To solve this problem, *Bontpart et al. (2020)* developed an affordable soil-based growth and imaging system which is large enough (approximately 6,000 cm$^2$) to allow vertical root growth. Although the above methods have high resolution, their throughput is relatively low. Therefore, *Treurnicht, Pagel & Esler (2015)* developed a novel phenotyping system, GROWSCREENRhizo, that can image roots at a throughput of 60 rhizotrons per hour, as verified by analyzing the root system of two dicot and four monocot plant species. Other platforms based on soil culture include GLO-Roots (*Rubén et al., 2015*), GLO-Bot (*LaRue et al., 2021*), PhenoRoots (*Martins et al., 2020*), and WinRoots (*Zhang et al., 2021a*). These methods can be used to obtain pictures of naturally growing roots. However, analyzing datasets from pictures can be time consuming and labor intensive. Therefore, transparent soil was proposed (*Downie et al., 2012*). Transparent soil consists of a matrix of solid particles and a pore network containing liquid and air. *Ma et al. (2019)* created a transparent soil formed by the spherification of hydrogels of biopolymers that can support root growth and allow root phenotyping *in vivo via* photography and microscopy. Soybean roots grown in transparent soil medium have been shown to exhibit striking resemblance to those developed in the real soil. Admittedly, transparent soil still has many shortcomings; for example, the size of the root volume (20 cm × 20 cm × 20 cm) is limited due to the transparency and the mechanical properties of its components. Also, the surface chemistry of the transparent soil is significantly different from that of the real soil. However, the use of transparent soil still has a great potential in promoting quantitative root characterization *in situ* using high-resolution imaging if its shortcoming can be solved.

## 3D root phenotyping platform

Although 2D root phenotyping methods provide a great convenience for root studies, these 2D methods are inherently limited by the information available from a single point of view, which only provided a limited set of easily measurable root traits. Therefore, there has been increased interest in developing capacity towards 3D root phenotyping

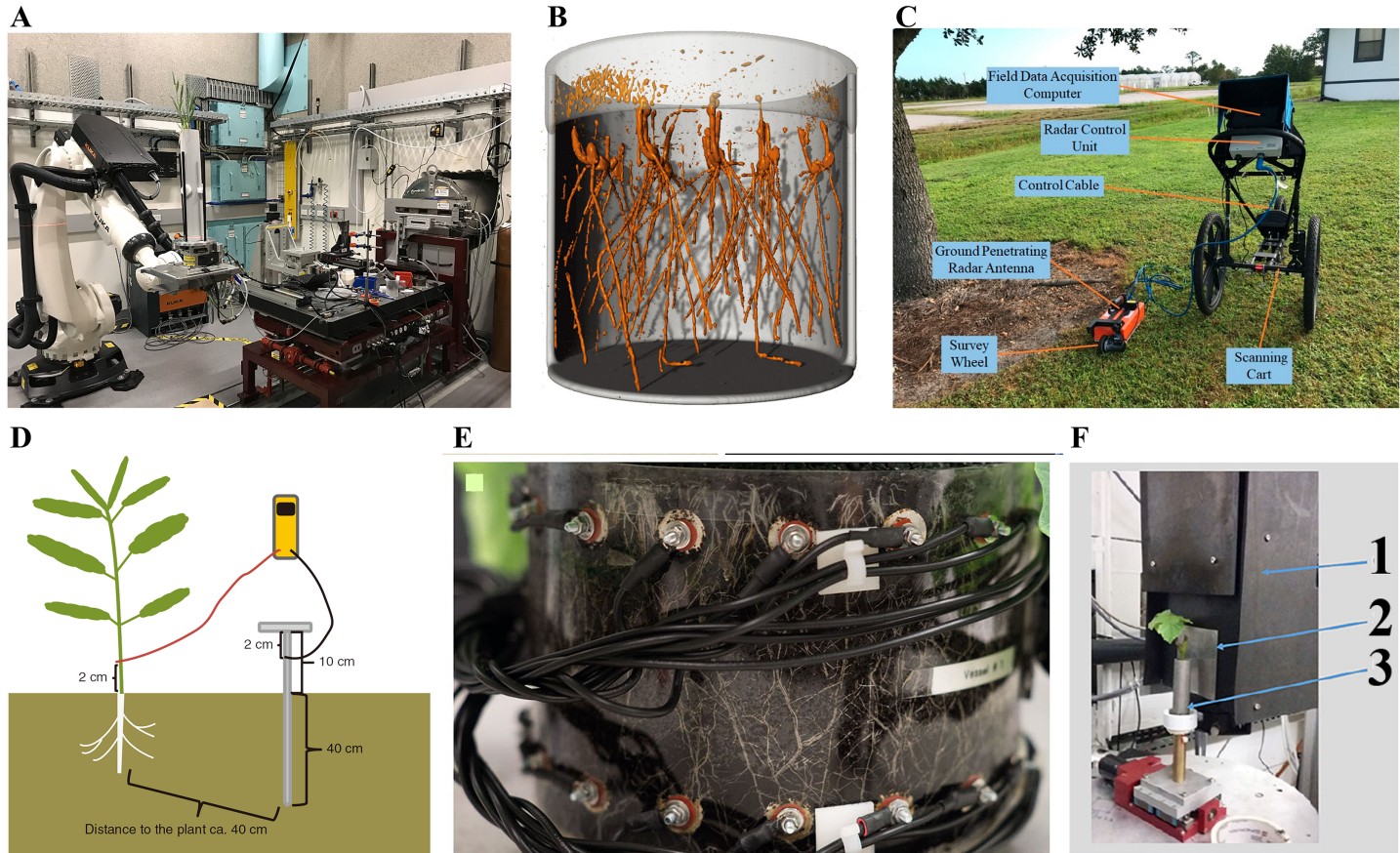

**Figure 2 Schematic representation of different 3D root phenotyping methods.** (A) Xray computed tomography (CT) (*Hou et al., 2022*); (B) magnetic resonance imaging (MRI) (*Daniel et al., 2022*); (C) ground penetrating radar (GPR) (*Zhang et al., 2019*); (D) electrical capacitance (EC) (*Schierholt et al., 2019*); (E) electrical impedance tomography (EIT) (*Corona et al., 2019*) (F) neutron tomography (NT) (*Krzyzaniak et al., 2021*). 1. CCD camera box; 2. scintillator; 3. container with plant.

technologies (Fig. 2), driven by technical advances and interdisciplinary approaches that allow digital reconstruction in 3D and high-throughput feature extraction.

X-ray computed tomography (CT) allows for the 3D reconstruction of root architecture in the soil (*Heeraman, Hopmans & Clausnitzer, 1997*). CT employs an X-ray beam from a source passing through the sample, which absorbs part of these beams *via* a process known as attenuation. The absorbed beams are recorded by a detector in series of 2D projections, which are further reconstructed into a 3D dataset. Material properties and electron density are the main factors influencing attenuation. The inner structure of samples becomes visible due to different densities and atomic numbers of the elements (*Plews, Atkinson & Mcgrane, 2009*; *Flavel et al., 2012*; *Metzner et al., 2015*). CT technology was first used in medicine and later applied in plant research 30 years ago (*Tollner, Verma & Cheshire, 1987*). However, resolution, scan time, and image segmentation have limited the large-scale application of CT in root phenotyping. Fortunately, recent advances in CT continue to facilitate its application in root phenotyping (*Mooney et al., 2012*). For example, *Teramoto et al. (2020)* visualized rice root architecture in 12 min (10 min for CT scanning and reconstruction and 2 min for image processing) using CT by applying

higher tube voltage and current and high-performance computing technology. This approach reduces the X-ray dosage to avoid adversely affecting rice growth (<0.09 Gy). In addition, it allows quantification of root architecture over time and in response to environmental stress by analyzing root 3D models derived from CT images. *Shao et al. (2021)* generated highly precise 3D models of maize root crowns *via* CT and created computations pipelines that could measure 71 features from each sample. *Herrero-Huerta et al. (2021)* developed a spatial-temporal root architecture modeling method based on CT, enabling the extraction of key root traits, including root number, length, angle, diameter, and volume of lateral roots. However, the application of CT technology is limited because it requires expensive equipment, and there are limits on the soil volume that can be scanned (*Morris et al., 2017*).

Magnetic resonance imaging (MRI) is another commonly used non-destructive 3D root phenotyping method. Living tissues have abundant magnetic moment of atomic nuclei, which can be manipulated using strong magnetic and radio-frequency fields to produce 3D datasets (*van Dusschoten et al., 2016*). MRI has been used to conduct root phenotyping in maize, bean, and barley (*Jahnke et al., 2009*; *Metzner et al., 2014*). The type of substrate and water content influences the MRI image quality (*Rogers & Bottomley, 1962*). For example, *Pflugfelder et al. (2017)* revealed that the thinner lateral roots (diameter <0.3 mm) of barely could still be resolved in five of the eight tested substrates, while, only the thicker roots were detectable in other substrates. Moisture above 70% of the maximal water holding capacity impedes MRI root for artificially composed substrates, however, for natural soil substrates, moisture in the range of 50%–80% of the maximal water holding capacity does not affect MRI root image quality. *Daniel et al. (2022)* recently analyzed the 3D root architecture of 288 winter wheat seedlings using a new workflow based on MRI, which can be categorized as medium-throughput phenotyping. Compared to X-ray CT, MRI has minor effects on plant growth because it does not utilize ionizing radiation. *Metzner et al. (2015)* compared CT and MRI by imaging roots growing in pots of three different sizes (the inner diameter were 34, 56, and 81 mm). CT showed more root details than MRI for the 34 mm diameter pot. In contrast, MRI detected more roots than CT in the 56 mm pot, suggesting that the effect of high water content is significantly greater on CT than on MRI. The hardware and software costs of installing MRI and CT are very high, and the equipment is difficult to relocate due to their large size (*Zappala et al., 2013*). In addition, MRI and CT technologies have been shown to restrict plant growth and development in a given container (*Poorter et al., 2012*). Collectively, these shortcomings limit the large-scale application of MRI and CT on root phenotyping.

Ground penetrating radar (GPR) is an emerging and rapidly evolving high throughput 3D root imaging method that is applicable in the field. GPR is a geophysical approach that detects shallow underground objects by emitting electromagnetic pulses. A portion of the pulses is reflected when it encounters a reflective surface. This progress is recorded as a function of travel time. Ultimately, these reflections can be quantified and generated into a 3D field, allowing for root visualization (*Alnuaimy et al., 2000*; *Jol, 2009*; *Liu et al., 2018*). GPR has been widely used to measure the coarse root (diameter >2 mm) of trees and shrub species, such as cassava (*Delgado et al., 2017*), loblolly pine (*Butnor et al., 2001*), elm

(*Li et al., 2013*), willow (*Li et al., 2015*), and citrus (*Zhang et al., 2019*). *Liu et al. (2018)* scanned winter and cane roots using GPR (1,600 MHz) and found significant relations between GPR indices and root parameters, implying that GPR can be applied to phenotype crop roots. However, GPR has certain limitations: (1) Expensive equipment of GPR limits its application in detecting crop roots, and it needs to reduce equipment coast in the future. (2) GPR signal can be affected by soil conditions which reduce the energy returned to the receiving antenna, resulting in inaccurate estimations of root (*Villordon, Ginzberg & Firon, 2014*). Future studies should address the problem by using newer antennas and incorporating data like soil pre-planting analysis.

Electrical capacitance (EC) is another 3D root imaging method applicable in the field. It uses a low-frequency alternating current (mostly less than 1 kHz) between the base of plant stem and the surrounding soil and then measures the resulting dielectric properties to re-establish the root system (*Chloupek, 1972*; *Dalton, 1995*). EC has been applied to phenotype roots of various crops, including soybean (*Cseresnyés et al., 2017*), maize (*Imre et al., 2018*), and wheat (*Cseresnyés et al., 2021*). However, the feasibility of the capacitance method has not been verified. Notably, some studies have reported that EC can be used to obtain reliable data on root phenotype. For example, *Cseresnyés et al. (2021)* found high correlation between root electrical capacitance and root dry mass of surface area by plant harvest method. Nevertheless, some studies have revealed inconsistencies in the results obtained using EC, casting doubt on the feasibility of the method. *Dalton (1995)* found that capacitance does not change significantly when the root system is cut off. Indeed, the EC method has many limitations. Specifically, EC requires the roots to be in contact with the soil solution to avoid underestimating the root traits (*Aulen & Shipley, 2012*). Also, the influence of other factors such as root density and physiological maturity on EC is still poorly understood.

In addition to the commonly used 3D root phenotyping methods, other 3D based methods, including electrical resistivity tomography (ERT), electrical impedance tomography (EIT), neutron radiography (NR), positron emission tomography (PET), thermoacoustic tomography (TT), electrical current source density (ECSD), and neutron tomography (NT) have been developed. The theory behind ERT contradicts that of EC. ERT generates high-resolution measurements by determining resistivity and further converts the measurement into a 3D model (*Pinheiro, Loh & Dickin, 1998*; *Atkinson et al., 2018*). Like GPR, ERT is mainly used for plants with large diameters like trees (*Rossi et al., 2011*; *Paglis, 2013*). However, EIT has also been applied to characterize the root phenotypes of corn and sorghum (*Srayeddin & Doussan, 2009*). EIT is based on the same theory as ERT, except it injects an alternating current rather than a direct current, which is superior in discriminating between roots and soil, thus can be used to depict plant-soil interaction (*Maximilian & Andreas, 2017*; *Mary et al., 2017*). *Corona et al. (2019)* visualized the root development of oilseed rape using EIT and demonstrated that EIT has the potential of becoming a low-cost tool for root phenotyping. NR is an imaging method that complements X-ray CT. Like X-ray CT, NR requires a beam; however, NR interacts with the nuclei instead of the electron shell. A primary advantage of NR method is the capacity to simultaneously monitor water distribution and root characteristics (*Menon*

*et al., 2007*; *Oswald et al., 2008*; *Leitner et al., 2014*). PET reconstructs a 3D image by detecting the distribution of γ (gamma) rays from short half-life radioactive tracers (*Atkinson et al., 2018*). $_{14}$C is the most used tracer (*Garbout et al., 2012*). However, the resolution of PET does not go beyond 1.4 mm, although it has a high sensitivity for tracers. Therefore, PET is usually combined with other tomographic techniques like MRI and X-ray CT to improve detection. *Garbout et al. (2012)* demonstrated the simultaneous use of PET and X-ray CT to image fodder radish root in sand. *Jahnke et al. (2009)* investigated root/shoot systems of sugar beet, radish, and maize growing in soil or sand by combining PET and MRI. TT is a safe, low-power, and cost-effective imaging technique with 300 μm resolution based on applying specific design of near field radio frequency applicators (*Aliroteh & Arbabian, 2017*). The ECSD approach was developed by *Peruzzo et al. (2020)*. The method involves applying a current from the plant to the soil and imaging the distribution and intensity of the electric current in the root-soil system. ECSD was further validated using rhizotron laboratory experiments on cotton and maize. NT can record root traits in soil filled growth container using a nuclear reactor or a high-energy particle accelerator (*Moradi et al., 2011*). The NT method has been applied to root phenotyping of maize (*Ali et al., 2018*) and grapevine (*Krzyzaniak et al., 2021*). Compared with the above-motioned complex 3D root imaging methods, *Clark et al. (2011)* developed the most simple and high-throughput 3D root phenotyping method. They grew two rice genotypes seedlings in a transparent gellan gum system attached to a digital camera for imaging and reconstructed and analyzed 3D root images using RootReader3D.

## Root image processing software

Recent improvements in root phenotyping methods and platforms have made it comparatively easy to obtain various large and high-quality images detailing the dynamics of the root system. Therefore, developing convenient and high-throughput software tools that can conduct objective, quantitative analyses of the root images is crucial. Hundreds of root image analysis software have been reported so far. The software can be divided into 2D and 3D root image processing software (Table 2).

2D root image processing software can be further divided into manual, semi-automated, and automated software based on its level of automation. Manual software is relatively rare because they are time consuming, subjective, and error-prone. WinRHIZO$^{TM}$ is one of the most widely used manual root analysis software. It can be used to analyze images coming from minirhizotron underground video camera systems or other sources that do not always offer a good contrast between roots and their background (*Arsenault et al., 1995*). Taking measurements using WinRHIZO$^{TM}$ involves manual tracing of the roots over the image using the mouse. WinRHIZO$^{TM}$ can obtain root morphological traits, such as root length, projected area, surface area, volume, and number of tips per diameter class, and topology structure, like branching angle and root system altitude. DART is a manual freeware based on human vision written in JAVA (*Bot et al., 2010*). DART can study root architecture and produce structure and flexible datasets of individual root dynamic parameters. It relies on manual manipulation to minimize the probability of

**Table 2 Overview of currently available root image analysis software.**

| Automation level | Software | Background | Dimension | Root trait | Advantege | Throughput | Species | Release time | Availability | Download | Reference |
|---|---|---|---|---|---|---|---|---|---|---|---|
| Manual | DART | Acetate sheet | 2D | Length/Branching order/Densities | Analysis of entire and complex root systems/Keep track of root colour | Medium | Quercus pubescens L./Solanum lycopersicum | 2010 | Free | http://www.avignon.inra.fr/psh/outils/dart | Bot et al. (2010) |
| | WinRHIZO™ | Soil | 2D | More than 20 traits | Root lifespan analysis | Low | Unlimited | 1995 | Paid | | Arsenault et al. (1995) |
| Semi-automated | EZ-Rhizo | Agar | 2D | 15 traits | Suitable for investigating a wide range of biological questions | High | Arabidopsis thaliana | 2009 | Free | http://www-ez-rhizo.psrg.org.uk | Armengaud et al. (2009) |
| | GiA Roots | Water | 2D | 19 traits | Add on new algorithms and trait estimation steps using plugins | High | Oryza sativa | 2012 | Free | https://www.quantitative-plant.org/software/giaroots | Galkovskyi et al. (2012) |
| | GLO-RIA | Soil | 2D | More than 10 traits | Relate root system parameters to local root-associated variables | Medium | Arabidopsis thaliana | 2015 | Free | https://github.com/rr-lab/GLO-Roots/tree/master/gloria | Rubén et al. (2015) |
| | GrowScreen-Root | Agar | 2D | Length of main and lateral roots/Number of lateral roots/Branching angle | Quantify complex root systems at a high throughput | High | Zea mays | 2009 | On-demand | | Nagel et al. (2009) |
| | Growth Explorer | Paper | 2D | Velocity-profile | Addresses both overall growth and local growth zones of roots | High | Cicer arietinum L./Phaseolus vulgaris L. | 2012 | Free | http://home.iitk.ac.in/~apal/growthexplorer.html | Basu et al. (2007) |
| | KineRoot | Paper | 2D | Spatio-temporal patterns/Curvature/Gravitropic | Generate reliable root growth data even in regions where there are very low contrast patterns | Medium | Phaseolus vulgaris/Arabidopsis thaliana | 2007 | On-demand | | Basu et al. (2007) |
| | MyROOT | Agar | 2D | Length | Recognize hypocotyls of different ages and morphologies | High | Arabidopsis thaliana | 2018 | Free | https://www.cragenomica.es/research-groups/brassinosteroid-signaling-in-plant-development | Betegón-Putze et al. (2018) |
| | rhizoTrak | Soil | 2D | More than 20 traits | Time-series annotation | Medium | Unlimited | 2019 | Free | https://github.com/prbio-hub/rhizoTrak | Möller et al. (2019) |

(Continued)

| Automation level | Software | Background | Dimension | Root trait | Advantege | Throughput | Species | Release time | Availability | Download | Reference |
|---|---|---|---|---|---|---|---|---|---|---|---|
| | RootNav | Paper/Agar/Water | 2D | More than 10 traits | Reconstruction and quantification of complex root architectures | High | Triticum aestivum/Arabidopsis thaliana/Brassica napus/Oryza sativa | 2013 | Open source | https://sourceforge.net/projects/rootnav/ | Pound et al. (2013) |
| | RootReader2D | Paper/Agar/Water | 2D | More than 10 traits | Measure individual roots from older or more highly overlapped root systems | High | Oryza sativa/Zea mays/Arabidopsis thaliana | 2013 | Free | http://www.plantmineralnutrition.net/ | Clark et al. (2013) |
| | RootScape | Agar | 2D | More than 10 traits | Rapidly and accurately characterize RSA variation in different genetic backgrounds or treatments | High | Arabidopsis thaliana | 2013 | Free | http://cmpdartsvr1.cmp.uea.ac.uk/wiki/BanghamLab/index.php/Software | Ristova et al. (2013) |
| | RootTip Trace | Agar | 2D | Length/Growth rate | Identify root tip | High | Arabidopsis thaliana | 2013 | Free | https://dinnenylab.info/ | Geng et al. (2013) |
| | RooTrak | Soil | 3D | 3D-reconstruction | Minimal user interaction/Adapt to changing root density estimates | High | Unlimited | 2011 | Free | https://www.nottingham.ac.uk/research/groups/cvl/software/rootrak.aspx | Mairhofer et al. (2012) |
| | SmartRoot | Transparent plate | 2D | More than 10 traits | Time-series handlin/Sampling-based analys/Vector-based representation of root | Medium | Lupinus albus/Zea mays | 2011 | Free | https://smartroot.github.io/ | Lobet, Pagès & Draye (2011) |
| Automated | Aria | Water | 2D/3D | 27 traits | Fast/Batch analysis/Ability to analyze 3D images | High | Zea mays | 2014 | Free | https://www.me.iastate.edu/bglab/software/ | Pace et al. (2014) |
| | ARTT | Paper/Gel | 2D | Root tip kinematics | Kinematic analysis | High | Zea mays/Oryza sativa. | 2013 | On-demand | | Russino et al. (2013) |
| | BRAT | Agar | 2D | 16 traits | Robust toward various experimental conditions | High | Arabidopsis thaliana | 2014 | Free | http://www.gmi.oeaw.ac.at/researchgroups/wolfgang-busch/resources/brat | Slovak et al., 2014 |
| | DIRT | Black imaging board | 2D | More than 70 traits | Automatic extraction of many root traits in a high-throughput fashion | High | Zea mays/Vigna unguiculata | 2014 | Free | http://dirt.iplantcollaborative.org/ | Bucksch et al. (2014) Das et al. (2015) |

| Automation level | Software | Background | Dimension | Root trait | Advantage | Throughput | Species | Release time | Availability | Download | Reference |
|---|---|---|---|---|---|---|---|---|---|---|---|
| | ElonSim | Agar | 2D/3D | Length | Processing of 3D images | High | Medicago truncatula/Rape/Sugar beet/Wheat | 2014 | Free | https://www.quantitative-plant.org/software/elonsim | Benoit (2014) |
| | EZ-Root-VIS | Agar | 2D | 16 traits | Capture RSA features of many individual plants/Visualize averaged RSAs for different genotypes under various environments or at different time points | High | Arabidopsis thaliana | 2018 | Free | http://www.psrg.org.uk/Rhizo-II.htm | Shahzad et al. (2018) |
| | faRIA | Soil/Agar | 2D | More than 10 traits | Without manual interaction with data and/or parameter tuning | High | Zea mays/Oryza sativa. | 2021 | Free | https://ag-ba.ipk-gatersleben.de/faria.html | Narisetti et al. (2021) |
| | GROW Map-Root | Black plastic | 2D | Root tip growth velocity | High spatial and temporal resolution | High | Zea mays | 2002 | On-demand | | Walter et al. (2002) |
| | IJ-Rhizo | Water | 2D | Diameter/Length | Carry out automated measurement of scanned images of root samples without sacrificing accuracy | Medium | Grape | 2013 | Open-source | https://www.quantitative-plant.org/software/IJ_Rhizo | Pierret et al. (2013) |
| | RNQS | Dark felt | 2D | Count/Length/Nodules | Standardized spatial analysis of nodulation patterns | Medium | Pisum sativum | 2014 | Free | http://hdl.handle.net/10393/30321 | Remmler et al. (2014) |
| | RootGraph | Water | 2D | Count/Length/Diameter | Image adaptation and graph optimization/Does not rely on any statistical learning | High | Hordeum vulgare/Triticum aestivum | 2015 | Free | https://www.quantitative-plant.org/software/RootGraph | Cai et al. (2015) |

(Continued)

| Automation level | Software | Background | Dimension | Root trait | Advantage | Throughput | Species | Release time | Availability | Download | Reference |
|---|---|---|---|---|---|---|---|---|---|---|---|
| | Root System Analyzer | Sandy soil | 2D | 18 traits | Distinguish root overlaps from branches | High | *Lupinus albus* | 2014 | Free | https://www.csc.univie.ac.at/apart/ | *Leitner et al. (2014)* |
| | RootFlowRT | Petri dish | 2D | Growth/Velocity-profile | Combination of optical flow methods | High | *Lycopersicon lycopersicum/ Lactuca sativa/ Aurinia saxatilis/ Phleum pratense* | 2003 | Free | http://www.bio.umass.edu/biology/baskin/ | *van der Weele et al. (2003)* |
| | RootFly | Soil | 2D | Color/Diameter/Length | Time savings over traditional manual analysis | High | Sweetbay magnolia/Freeman maple | 2008 | Free | https://cecas.clemson.edu/~stb/rootfly/ | *Zeng, Birchfield & Wells (2008)* |
| | RootReader3D | Gellan gum | 3D | 27 Ttraits | Automated and interactive features | High | *Oryza sativa* | 2011 | Open source | http://www.plantmineralnutrition.net | *Clark et al. (2011)* |
| | RootTrace | Agar | 2D | Length/Curvature/Stimulus response parameters | Process long time-lapse sequences | High | *Arabidopsis thaliana* | 2009 | Open source | https://www.nottingham.ac.uk/research/groups/cvl/software/roottrace.aspx | *French et al. (2009)* |
| | RhizoVision Explorer | Transparent plate/Water | 2D | More than 20 traits | Default broken roots mode | High | Unlimited | 2021 | Open source | https://doi.org/10.5281/zenodo.3747697 | *Seethepalli et al. (2021)* |
| | RSAtrace3D | Soil | 3D | Length/Root growth angle/Root distribution parameters | High expandability of the vectorization and phenotyping algorithm | Medium | *Oryza sativa* | 2021 | Open source | https://rootomics.dna.affrc.go.jp/en/ | *Teramoto, Tanabata & Uga (2021)* |

mistakes and biases in datasets. The advantage of manual software is that it can be used to analyze the lifespan of roots by keeping track of root color manually.

Currently, there are many semi-automated root analysis software, including EZ-Rhizo (*Armengaud et al., 2009*), GrowScreen-Root (*Nagel et al., 2009*), GiA Roots (*Galkovskyi et al., 2012*), GLO-RIA (*Rubén et al., 2015*), KineRoot (*Basu et al., 2007*), MyROOT (*Betegón-Putze et al., 2018*), Multi-ADAPT (*Ishikawa & Evans, 2010*), RootNav (*Pound et al., 2013*), RootReader2D (*Clark et al., 2013*), RootScape (*Ristova et al., 2013*), RootTipTrace (*Geng et al., 2013*), and SmartRoot (*Lobet, Pagès & Draye, 2011*). EZ-Rhizo is a Windows-integrated and semi-automated computer program that can be employed to quantify multiple root parameters of plants growing on agar medium. The software entails following four pre-defined operations after opening an image, *i.e.*, make the image black and white, remove box, remove noise, and dilate. After that, the following five operations are used to quantitatively analyze root traits, *i.e.*, skeletonize, re-touch, find roots, confirm roots, and save experiment (*Armengaud et al., 2009*). RootNav is a widely used free and open source root image analysis software that allows semi-automated quantification of complex root traits in various plant species and images (*Pound et al., 2013*). RootNav takes a top-down approach and utilizes the expectation-maximization (EM) clustering algorithm (*Dempster, 1977*) to calculate the likelihood that a given pixel corresponds to roots. Then these likelihood values are used to estimate each pixel that effectively fits a model of individual root. Regarding accuracy, RootNav has been evaluated on winter wheat, *Brassica napus*, and rice. The root length measured by RootNav has been found to be 2% shorter than those measured by manual methods; however, RootNav is faster and easier to use than manual methods. Notably, RootNav was recently upgraded to RootNav 2.0 based on extremely deep multi-task Convolutional Neural Network architecture (*Robail et al., 2019*). KineRoot is an earlier application of automated root analysis software developed by Matlab 7.0 (*Basu et al., 2007*). KineRoot analyzes root image by following two basic steps. First, the marker pointers on the root image are tracked using three search algorithms, and then, the root edges are identified automatically by an edge detection algorithm. KineRoot can analyze many images to generate local root growth and root curvature data quickly, allowing kinematic analysis of root growth and gravitropic responses for various root types. The main advantage of the KineRoot software is that it can detect root edges and measure curvature and elongation rates of roots. However, KineRoot can only be used to analyze microscope scale images. Also, only a limited, number of roots can be analyzed at each step. SmartRoot is an operating system-independent freeware based on ImageJ, which combines a powerful tracing algorithm and a root vectorial representation (*Lobet, Pagès & Draye, 2011*). The advantage of SmartRoot is that it can be used to analyze low quality images as long as the roots reach two to four pixels wide. However, SmartRoot is not suited for high-throughput analysis because its design allows substantial amount of user interference (*Marié et al., 2014*). RootReader2D is a semi-automated analysis software based on Java programming language (*Clark et al., 2013*). RootReader2D is free and publicly available. The program integrates user-guided features and batch processing functionality, increasing flexibility and enhancing efficiency when measuring root growth traits from specific roots or entire

root systems during large-scale phenotyping studies. RootReader2D can be used to analyze root images in various culture environments, such as hydroponics, gels, paper pouches, and soil bases.

Similar to semi-automated software, several automated root analysis software have been developed, including ARIA (*Pace et al., 2014*), EZ-Root-VIS (*Shahzad et al., 2018*), HYPOTrace (*Wang et al., 2009*), RhizoVision Explorer (*Seethepalli et al., 2021*), Root System Analyzer (*Leitner et al., 2014*), RootGraph (*Cai et al., 2015*), and RootTrace (*French et al., 2009*). RootTrace is a high-throughput tool previously used to analyze the roots of Arabidopsis seedling grown on agarose plates. It is based on top-down approach (*French et al., 2009*) and employs automatic tracking techniques to track roots from a user-defined start location. It also uses a condensation method (*Isard & Blake, 1998*) to track down the root until the root tips are detected. The top-down approach is robust to all kinds of noise effects and is quite flexible across different image sets. RootTrace requires minimal interaction from the user, permitting long time-lapse sequences processing. However, it still needs a user interaction on the first frame. ARIA captures multiple root traits from images of seedling roots by converting the images into an equivalent graph (*Pace et al., 2014*). This process is done by labeling each root image pixel into a vertex and linking nearest neighbor pixels with edges. ARIA can rapidly extract data (within approximately 20 s) profits from a friendly user GUI interface. In addition, ARIA can be used to analyze most standard image formats and has been demonstrated to support accurate measurements by comparing 27 traits measured results with WinRhizo Pro 9.0. ARIA (ARIA 2.0) was recently applied to study soybean root phenotype and achieved good results (*Falk et al., 2020*). RootGraph is the first tool to use a weighted graph optimization process to produce a fully automatic and robust method for detailed description of root traits (*Cai et al., 2015*). RootGraph begins by distinguishing primary roots from lateral roots, then comprehensively quantifying root traits for each identified primary and lateral root, and finally combining lateral root features with the specific primary root traits from which the laterals emerge. RootGraph has been verified to be accurate, robust, and high-throughput by comparing it with other automated and semi-automated software, and manual measurements. Furthermore, RootGraph utilizes image adaptation and graph optimization instead of statistical learning. It can also remove any noise caused by soil particulates remaining after cleaning roots. GLO-RIA is an Image J plugin consisting of two modules that allow automated measurement of numerous root traits using a combination of existing tools (*Rubén et al., 2015*). GLO-RIA can also relate root trait parameters to local root-associated variables such as reporter expression intensity and water content in soil. The first module performs four different types of root system analysis, which are fully automated by default, but can be adjusted manually if needed. The second module analyzes multi-layered images, including combinations of reporter gene, root structure, and soil moisture through five different types of analysis. *Seethepalli et al. (2021)* developed an open-source, fast image processing, and reliable measurement software called RhizoVision Explorer. RhizoVision Explorer is mainly used to analyze root images obtained by a flatbed scanner from pots or soil cores after washing. RhizoVision Explorer was successfully validated by comparing its analysis results with

those of WinRhizo™ and IJ_Rhizo using a simulated root image set, which generally showed consistent results. RhizoVision Explorer facilitates the standardization of root traits and morphological measures by a user-friendly, fast, generalist, collectively improvable design. Future improvements of RhizoVision Explorer should include incorporating powerful topology analysis to predict root order, diameter, and angle.

Automatic analysis methods based on convolutional neural networks (CNNs) have also developed rapidly in recent years. CNNs can directly extract target traits from an input image by combining deep learning and computer vision technology (*Lecun, Bengio & Hinton, 2015*). *Tao et al. (2019)* developed a fully automated tool based on CNNs called SegRoot which can extract roots from complex soil backgrounds. Meanwhile, a quantified metric (the dice score) was used to assess the qualitative segmentation performance. A high degree of correlation was achieved ($R^2$ = 0.9791) by comparing the root length obtained by SegRoot *vs.* human traced. However, SegRoot has been shown to underestimate root length because it can miss fine roots and the existence of blurred areas. Similarly, *Shen et al. (2020)* developed an automated image segmentation software based on the DeepLabv3+ CNNs and achieved excellent results. Nevertheless, getting researchers without an in-depth knowledge of machine learning to use this method proficiently remains a challenge. To address this limitation, *Han et al. (2021)* developed an AI-based software called RootPainter, which uses a modified U-Net architecture (*Ronneberger, Fischer & Brox, 2015*) equipped with an interface for corrective annotation for easy use. The automated segmentation method based on CNNs will revolutionize the measurement of plant roots in soil.

Although 3D root phenotyping methods have continued to advance rapidly, the development of corresponding image analysis tools has lagged. The main reason is that extracting 3D root system parameters entails interpreting the number of image pixels, color grade and size. It also involves constructing a spatial distribution function, which greatly increases the difficulty of the software design. iRoCS Toolbox (*Schmidt et al., 2014*), RootReader3D (*Clark et al., 2011*), RooTrak (*Mairhofer et al., 2012*), and NMRooting (*van Dusschoten et al., 2016*) are the most used 3D root phenotyping analysis software. iRoCS Toolbox is an open-source software package that enables direct and quantitative analysis of the root tips at cellular resolution (*Schmidt et al., 2014*). iRoCS Toolbox groups the nuclei/cells into root tissue layers by detecting nuclei or segment cells and automatically fits the coordinate system. All processes are performed automatically except for marking the quiescent center. iRoCS Toolbox enables researchers to rapidly standardize their data within a single framework and quantitatively compare root cohorts. iRoCS Toolbox drastically reduces the time required to fully annotate a single root by associating algorithmic pipelines to automatically recognize cell boundaries and nuclei. The time saved increases the number of roots that can be annotated, ensuring impartial evaluation of previously hidden and mild developmental phenotypes and making statistical analyses possible. RootReader3D (*Clark et al., 2011*) is a custom-designed software that utilizes a silhouette-based back-projection algorithm combined with cross-sectional volume segmentation to generate 3D root models (*Mulayim, Yilmaz & Atalay, 2003*; *Zhu et al., 2006*). RootReader3D integrates multifarious viewing interfaces

and mouse and keyboard commands to support visualizing and interacting with the 3D roots reconstructions. RootReader3D measurements are validated by comparing them with 2D measurements. However, this software is only suitable for analyzing root images with a single background because it cannot eliminate the influence of non-root substances in the images. RooTrak is an automatic software used to analyze images generated by X-ray CT using the top-down approach (*Mairhofer et al., 2012*). RooTrak views three-dimension CT data as a series of x-y cross-sectional images aligned along the z-axis. Root cross sections move around the image following the image stack traversed, reflecting the shape of the scanned root. RooTrak can obtain a range of root traits from various plant species grown in multifarious contrasting soil with minimal user intervention, a feature that will facilitate future root phenotyping efforts. NMRooting is an automated analysis software for analyzing MRI datasets written in Python (*van Dusschoten et al., 2016*). NMRooting achieves 3D visualization through Mayavi (*Ramachandran & Varoquaux, 2011*). *Teramoto, Tanabata & Uga (2021)* recently developed RSAtrace3D, a robust 3D root architecture vectorization software for monocot root phenotyping. RSAtrace3D implements graphical user interface by Python and can be applied to analyze rice X-ray CT images and various 3D images of other monocots.

## SUMMARY AND PERSPECTIVES

The current review focuses on recent advances in *in-situ* root phenotyping tools. The next challenge is to apply these phenotyping platforms in large-scale quantitative genetic analysis. The challenges require interdisciplinary efforts, from mathematics to computer science to root biology, and applied fields, including crop breeding and agronomy. Root biology and root-soil interaction, including the soil microbiome, spans multiple spatiotemporal scales and disciplines and is extremely complex. Therefore, root phenotyping should be extended to the rhizosphere phenotype, defined as root and root-influenced soil describing 'the manifestation of a plant's genetics' in the soil (*York et al., 2016*). Rhizosphere phenotyping greatly increases the opportunity of discovering new phenotypes related to root function, such as the rhizosheath traits and their association with root hairs. Mobile, easy-to-build cross-lab reproducible test systems will be new frontiers for future root and rhizosphere phenotyping studies. These innovative technologies and platforms are collectively driving the selection of the next generation of crops to address existing global food security challenges.

### Funding
This research was funded by the National Natural Science Foundation of China, grant number 31871569. The funders had no role in study design, data collection and analysis, decision to publish, or preparation of the manuscript.

## Grant Disclosures

The following grant information was disclosed by the authors:
National Natural Science Foundation of China Grant Number: 31871569.

## Competing Interests

The authors declare that they have no competing interests.

## Author Contributions

- Anchang Li performed the experiments, analyzed the data, prepared figures and/or tables, and approved the final draft.
- Lingxiao Zhu performed the experiments, prepared figures and/or tables, and approved the final draft.
- Wenjun Xu analyzed the data, authored or reviewed drafts of the article, and approved the final draft.
- Liantao Liu conceived and designed the experiments, authored or reviewed drafts of the article, and approved the final draft.
- Guifa Teng conceived and designed the experiments, authored or reviewed drafts of the article, and approved the final draft.

## Data Availability

The manuscript is a literature review and there is no raw data.

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
