# Peer review of "Recent advances in methods for in situ root phenotyping"

_PeerJ, doi:10.7717/peerj.13638_

## Round 0.1 · original submission · Minor Revisions

The reviewers have found the manuscript to be of significant importance. However, it needs minor corrections before acceptance.

Reviewer 1 has suggested that you cite specific references. You are welcome to add it/them if you believe they are relevant. However, you are not required to include these citations, and if you do not include them, this will not influence my decision.

·

Basic reporting

Comments:
The hidden half of crop plant imparts much important to researchers. Due to difficulties in phenotyping much of the researchers were less interested on its study. Low repeatability and reproducibility of root data is limitation. High plasticity of root traits and GXE interaction limits its application in breeding program. Despite quantitative traits, still this manuscript might help the root phenomics scientists accustomed with different root system analysis, its advantage, accuracy of data and also software tools available (freely or paid).
- While going through the manuscript, the comments has been mentioned on manuscript itself. Pl go through comments and incorporate accordingly.
- If possible, kindly insert image of some of the tools to make manuscript of good quality.
- Kindly add, mobile base application that can analyze root traits in a paragraph.
- In case of software for root analysis, whether scanning based or is of image based.
- Kindly add some of the reference related to non destructive, forest tree root analysis software (Biovis software).
Some important reference might add flavor to this review:
Tracy, S. R., Nagel, K. A., Postma, J. A., Fassbender, H., Wasson, A., & Watt, M. (2020). Crop improvement from phenotyping roots: highlights reveal expanding opportunities. Trends in plant science, 25(1), 105-118.
Das, A., Schneider, H., Burridge, J., Ascanio, A. K. M., Wojciechowski, T., Topp, C. N., ... & Bucksch, A. (2015). Digital imaging of root traits (DIRT): a high-throughput computing and collaboration platform for field-based root phenomics. Plant methods, 11(1), 1-12.
Pioneer work of Foulke et al is missing.
Reference of Rumesh Ranjan on root traits is missing.

Experimental design

NA

Validity of the findings

NA

Reviewer 2 ·

Basic reporting

The article is well written, describing very well about the . The description of the root insitu phenotyping methods and image processing softwares is good. However, information on measurable root traits under each method is to be desired. I have attached pdf with comments for revision

Experimental design

No

Validity of the findings

No

Additional comments

No

Annotated reviews are not available for download in order to protect the identity of reviewers who chose to remain anonymous.

Reviewer 3 ·

Basic reporting

The review summarizes a large body of literature on root phenotyping and related image analysis methods. The Introduction adequately introduces the subject matter.

While very few reviews if any are available that focus on both root survey and image analysis methods, I believe there are some very good reviews addressing root phenotyping, or addressing root imaging analysis. I suggest authors to briefly discuss about this in the Introduction.

Experimental design

The study design appears to be comprehensive and unbiased, and the review contents are organized logically.

Validity of the findings

The review provides a good summary of current literature on root phenotyping and imaging, including a discussion of pros and cons of numerous methods. Some of these comments may be based on authors' experience, while others may be based on literature survey; in the latter case, I see that proper citations are provided. So, I think the discussion is useful for readers.

Additional comments

I have only a few minor comments:

(1) Lines 37-38: Please provide a proper citation of the source of this statement. When was this calculation done and what assumptions were made when making this statement? Why has this been almost accepted as a fact?

(2) Lines 70-71: This disadvantage is too superficial that it is applicable to any other root survey method.

(3) Line 78-79: I don't agree. I think some of these methods for sure can be, and have been, used to examine dynamic changes of roots, even though they are cumbersome and inefficient.

(4) Lines 587-497; Since this is a review, there should not be an "experiment" to design. You can say someone conceived the idea and designed the study, but not the experiment.

---

## Round 0.2 · accepted · Accept

All necessary corrections have been incorporated by the authors.